# Aging, Physical Exercise, Telomeres, and Sarcopenia: A Narrative Review

**DOI:** 10.3390/biomedicines11020598

**Published:** 2023-02-17

**Authors:** David Hernández-Álvarez, Juana Rosado-Pérez, Graciela Gavia-García, Taide Laurita Arista-Ugalde, Itzen Aguiñiga-Sánchez, Edelmiro Santiago-Osorio, Víctor Manuel Mendoza-Núñez

**Affiliations:** 1Research Unit on Gerontology, FES Zaragoza, National Autonomous University of Mexico, Mexico City 09230, Mexico; 2Hematopoiesis and Leukemia Laboratory, Research Unit on Cell Differentiation and Cancer, FES Zaragoza, National Autonomous University of Mexico, Mexico City 09230, Mexico

**Keywords:** aging, sarcopenia, oxidative stress, telomeric length, repair mechanisms, exercise

## Abstract

Human aging is a gradual and adaptive process characterized by a decrease in the homeostatic response, leading to biochemical and molecular changes that are driven by hallmarks of aging, such as oxidative stress (OxS), chronic inflammation, and telomere shortening. One of the diseases associated with the hallmarks of aging, which has a great impact on functionality and quality of life, is sarcopenia. However, the relationship between telomere length, sarcopenia, and age-related mortality has not been extensively studied. Moderate physical exercise has been shown to have a positive effect on sarcopenia, decreasing OxS and inflammation, and inducing protective effects on telomeric DNA. This results in decreased DNA strand breaks, reduced OxS and IA, and activation of repair pathways. Higher levels of physical activity are associated with an apparent increase in telomere length. This review aims to present the current state of the art of knowledge on the effect of physical exercise on telomeric maintenance and activation of repair mechanisms in sarcopenia.

## 1. Introduction

Population aging has been called a “silver or gray tsunami”, considering the implications and challenges of the accelerated increase in the population of older adults in the world [1,2]. In this context, the World Health Organization (WHO) has reported the following data: “(i) Between 2015 and 2050, the proportion of the world’s population over 60 years will nearly double from 12% to 22%; (ii) by 2020, the number of people aged 60 years and older will outnumber children younger than 5 years; (iii) in 2050, 80% of older people will be living in low- and middle-income countries; (iv) the pace of population aging is much faster than in the past; and (v) all countries face major challenges to ensure that their health and social systems are ready to make the most of this demographic shift”. In this regard, the challenge implies the higher prevalence and incidence of chronic non-communicable diseases (NCDs) [3].

López-Otín et al. (2013) published a review on “the hallmarks of aging” proposing nine mechanisms as the biochemical and molecular processes that cause aging: (i) Genomic instability; (ii) telomere attrition; (iii) epigenetic alterations; (iv) loss of proteostasis; (v) deregulated nutrient-sensing; (vi) mitochondrial dysfunction; (vii) cellular senescence; (viii) stem cell exhaustion; and (ix) altered intercellular communication [4]. These biochemical and molecular alterations significantly increase the risk of presenting NCDs, among which sarcopenia stands out. Similarly, it has been shown that physical exercise training has an antioxidant and anti-inflammatory effect, and protects against telomere shortening during aging [5]. In this context, this review aims to present the current state-of-the-art of knowledge on the effect of physical exercise on telomeric maintenance and activation of repair mechanisms in sarcopenia.

## 2. Human Aging

There is no consensus on the definition of human aging. Among various attempts to define it, biological aging has received the most attention and is supported by numerous aging theories. It is estimated that there are over 300 of these theories [6]. Nevertheless, Troen (2003) points out five characteristics common to aging in mammals: (i) Increased mortality after adulthood. An exponential growth in mortality has been described with advancing age after the reproductive stage. (ii) Changes in tissue composition with age. Decrease in muscle and bone mass with aging, increase in adipose tissue, development of lipofuscin deposits, and crosslinking with structural proteins, such as collagen, due to processes that occur in aging, such as oxidation and glycosylation, and increased advanced glycation end products (AGEs). (iii) Progressive decrease in physiological capacity. It has been documented that there are many physiological changes with increasing age after adulthood, even in the absence of NCDs. (iv) Reduction in adaptation to environmental stimuli. With aging, there is a gradual decline in the ability to maintain homeostasis. (v) Increased vulnerability to disease [7]. Changes related to aging affect the cell function, leading to tissue and organ dysfunction, which ultimately triggers systemic diseases [8].

Similarly, Lemoine (2020) in a review and analysis of the definition of biological aging, has pointed out five characteristics commonly used in biogerontology to define aging: (i) Structural damage; (ii) functional decline; (iii) depletion of a reserve required to compensate for the decline; (iv) typical phenotypic changes or their cause; and (v) increase in the probability of death (or disease). Notably, any definition of aging must include some of these characteristics. Moreover, Cohen et al. (2020) demonstrated that there is no consensus on the definition of aging [9,10]. In this context, our research group has defined human aging as a “gradual and adaptive process characterized by a relative decrease in the homeostatic response, due to morphological, physiological, biochemical, psychological, and social factors, fostered by changes inherent to aging” [11].

It is estimated that by 2030, one in six people around the world will be 60 years or older [3]. Therefore, efficient socio-health care and the fostering of self-care are needed to improve the quality of life in this age group, since aging is associated with chronic-degenerative diseases that can occur simultaneously, increasing morbidity and ultimately death [4,12]. The deteriorated physiological function of the organism in aging is a risk factor for certain diseases, such as cancer, type 2 diabetes mellitus (T2DM), cardiovascular disorders, and neurodegenerative diseases, while in terms of skeletal muscle, the progressive loss of mass and strength could lead to sarcopenia [4,13].

## 3. Oxidative Stress

In aerobic and aerotolerant organisms, there is a balance between the production of reactive oxygen species (ROS) and the antioxidant defense system that maintains homeostasis, thus regulating the intracellular redox state. Therefore, oxidative stress (OxS) arises when there is an imbalance between the formation of oxidants and antioxidants [14,15,16]. 

OxS leads to structural lesions in lipids, proteins, and nucleic acids. Damage to the DNA is of great importance, since it serves as a permanent copy of the genome, and changes in its structure can impact the following generations [17,18,19].

Among the main damages generated in the DNA by ROS are the single-strand breaks (SSB) and double-strand breaks (DSB), as well as the formation of the adduct 8-hydroxy-2′-deoxyguanosine (8-OHdG) that causes the production of transversions (replacement of a purine with a pyrimidine or vice versa), which is highly related to different types of cancer. One of the DNA regions that present the greatest susceptibility to the suffering of damage by OxS is the telomeric region, due to its high content of guanine waste [20,21].

In this regard, guanine presents the lowest redox potential compared to the other DNA bases [22], and therefore, is more susceptible to OxS damage.

Similarly, the oxidation of lipids and proteins can generate intermediaries that could react with the DNA [23]. In the case of lipid decomposition, one of the final products of lipid peroxidation is the malondialdehyde (MDA), which forms the following adducts: Deoxyadenosine (M1dA), deoxycytidine (M1dC), and deoxyguanosine (M1dG) [24]. Meanwhile, as protein oxidation products, carbonyl groups, such as aldehydes and ketones, are generated [25].

### Oxidative Stress and Aging

The OxS theory of aging suggests that the functional decay associated with age is due to the progressive and irreversible accumulation of oxidative damage in biomolecules by reactive oxygen and nitrogen species (RONS) [26,27], which have a negative impact on the physiological function, development of diseases, and life expectancy [28].

Studies carried out in humans consider that age is a factor associated with the increase in OxS, evidenced by an increase in lipid peroxides, decrease in total antioxidant state in plasma, by the glutathione peroxidase (GPx) activity, and with no apparent changes in superoxide dismutase (SOD) levels in peripheral blood erythrocytes [29].

Similarly, studies have been carried out in mammals as well as in other species, and the findings obtained coincide, observing positive associations between the concentration of antioxidants and chronological life, while the amount of oxidative damage to the DNA is negatively associated with a useful life [30,31]. Similarly, two common characteristics have been identified in long-lived species: A low production of free radicals and a high repair rate of the damaged DNA [32].

Additionally, in aging experimental models, an increase in oxidized proteins in hepatocytes has been observed, which is possibly due to the fact that the proteases which degrade them have a lower activity, or are defective [33].

It has been proposed that the increase in OxS accelerates the development of certain pathologies with age (cardiovascular, renal, neurodegenerative, frailty, and sarcopenia), while the decrease in OxS retards them. This may be due to the fact that OxS exerts a very important role in the development of these diseases [26,34].

Both frailty and sarcopenia are closely related, since sarcopenia is an important component of frailty syndrome and both are considered predictors of morbidity, disability, and death in the elderly [35].

## 4. Sarcopenia

Sarcopenia is a progressive skeletal muscle disorder (essential for functions, such as postural support, breathing, thermogenesis, and movements), which is represented by the loss of mass and muscle strength [36,37], as well as a decrease in the number of motor units and myofibrillar components, denervation, and atrophy for disuse associated with age [38]. Some factors that can accelerate its progression are malnutrition, chronic diseases, and physical inactivity [39].

This geriatric syndrome is associated with a greater risk of falls, disability, physical frailty (strength, physiological function, and decreased resistance), functional deterioration, greater dependence, and/or mortality [40,41,42,43]. In addition, the reduction in the regenerative capacity of muscle fibers is due to the loss of satellite cells (muscle stem cells) [38]. Under normal conditions, these types of cells are activated by stimuli as growth signals that ensure homeostasis, in order to repair the integrity and function of fibers [36].

In 2016, sarcopenia was recognized as a disease [44]. In this context, the prevalence of this disease in the world is 10%, although it has been pointed out that it can reach up to 30% in people over 60 years [45,46]. Similarly, it is estimated that after the fifth decade, muscle mass decreases between 1% and 2% annually and strength decreases by 3% from 60 years of age. As a consequence of the denervation of motor units and conversion of type II muscle fibers, rapid contraction, which allows for calcium storage, facilitating nerve conduction in fibers type I, and slow contraction, which hydrolyzes ATP slowly to contract, cause alterations in muscle strength [47,48].

Generally, exercise interventions have been shown to significantly improve grip strength, knee extension strength, lower extremity muscle mass, gait speed, and functional mobility among older adults with sarcopenia [49]. Moreover, physical exercise coupled with an adequate diet are considered as the main roads to the prevention of sarcopenia.

On the other hand, the loss of skeletal muscle and the accumulation of intramuscular fat mass are associated with a wide variety of pathologies through a combination of factors, which include inactivity, chronic inflammation, mitochondrial dysfunction, insulin resistance, and OxS (Figure 1) [50].

### Oxidative Stress and Sarcopenia

In sarcopenia, underlying mechanisms, such as OxS and chronic inflammation, participate in age-related muscle atrophy. In the case of OxS, it is well established that the oxidation of mitochondrial proteins increases with age, and in turn, is negatively associated with muscle strength and malfunction of skeletal muscle [51].

In sarcopenic patients, an imbalance in the redox state has been documented, due to an increase in the oxidized glutathione ratio/reduced glutathione (GSSG/GSH) in peripheral blood and MDA/HNE adduct formation in plasma, which are both correlated with a greater predisposition to cardiovascular and cancer risk [52,53]. Similarly, they have been considered as possible mortality predictors through the decrease in pre-albumin, albumin, and transferrin, and increase in serum carbonylated proteins [54].

In regard to the antioxidant protection mechanisms, a decrease in catalase activity (CAT) has been evidenced with no apparent changes in the activity of SOD and GPx in plasma [55].

Advanced oxidation protein products (AOPP) and total radical trapping antioxidant parameter (TRAP) were associated with sarcopenia in chronic obstructive pulmonary disease (COPD) [56], whereas a decrease in glutathione reductase is associated with a risk of death and fewer days of survival in frail sarcopenic patients [57].

Meanwhile, in experimental models, the transverse area of myocytes and the weight of the muscle are reduced. In addition, the levels of the OxS markers, such as the 4-hydroxy-2-nonenal and 3-nitrotyrosine are higher in aged mice, compared to the young [58].

To date, two mitochondrial fractions have been identified that differ according to their location and the way they participate in the pathogenesis of sarcopenia. Subsarcolemmal mitochondria (SS) isolated from old muscle are attributed to the greater production of ROS, while intermyofibrillar mitochondria (IMF) are more likely to induce apoptosis under the stimuli of cell death and ROS, causing the atrophy of muscle fibers to be muscular [59,60]. Similarly, it has been documented that the decrease in the number of copies of mitochondrial DNA (mtDNA) in the aged skeletal muscle can be associated with sarcopenia. The dysfunction that arises from the oxidative damage to the mtDNA causes a decrease in important proteins for the transport of electrons, creating a vicious circle between ROS and mitochondrial deregulation, which finally leads to cell death by apoptosis [59,61,62,63]. For example, the levels of apoptotic markers increase during aging in normal mice, and the accumulation of mtDNA mutations may promote apoptosis, acting as a driving mechanism for aging. Increased cleavage of caspase-3, an enzyme responsible for the biochemical and morphological changes that occur in apoptotic cells, is a characteristic of skeletal muscle during normal aging and results in the loss of muscle fibers [64,65].

## 5. Inflammation

Inflammation is considered as a mechanism for protection against infections, tissue injuries, and microorganisms. These stressors can go unnoticed if they are not resolved by an efficient anti-inflammatory response, which can reverse the damage and restore homeostasis, known as acute inflammation. However, if inflammation is excessive, it can encourage homeostatic changes and be harmful, progressing to chronic inflammation [66,67].

Inflammatory markers are involved in various pathological states, such as cardiovascular and neurodegenerative diseases, cancer, T2DM, obesity, aging [67,68,69,70], sarcopenia, dementia, etc. This can lead to disability, frailty, and death. Some possible mechanisms associated with inflammation have been established, which are obesity, senescence, and OxS [71].

A characteristic of chronic inflammation is the increase in cytokines, which make the immune system work continuously but not in an effective way [72].

Anti-inflammatory cytokines (IL-4, IL-10) are immunoregulatory molecules that control the response of pro-inflammatory cytokines (IL-1, tumor-alpha necrosis factor (TNF-α) that act in synergy) [73,74].

Other homeostatic responses of the body against damage is the production of acute phase proteins, such as C-reactive protein (CRP), ceruloplasmin, fibrinogen, etc., that play an important role in both the acute and chronic inflammatory process. This type of protein is an effective tool to identify the intensity of the inflammatory process and is used for diagnostic purposes [75].

### 5.1. Inflammation and Sarcopenia

A mechanism underlying aging is the elevation of the surrounding levels of pro-inflammatory cytokines, known as chronic low-grade inflammation, where there is an increase in the TNF-α factor, interleukin-1β (IL-1β), and interleukin 6 (IL-6) [76,77,78].

This type of inflammation favors the loss of strength, muscle mass, functionality, anabolic resistance (decreased sensitivity of skeletal muscle before anabolic stimuli) [79,80], and cellular senescence, with the acquisition of senescence-associated secretory phenotype (SASP) being associated with frailty [81]. There is strong evidence that links sarcopenia to inflammation, due to the increase in biomarkers, such as adiponectin, CRP, and erythrocyte sedimentation rate [81,82,83].

Additionally, in a meta-analysis, the increase in CRP was evidenced, without apparent change in TNF-α and IL-6 [84]. There are contradictory data regarding these last two cytokines, since other studies indicate that patients with sarcopenic obesity (simultaneous presence of sarcopenia and excess adiposity) show high levels of IL-6 and additional inflammatory loads with the presence of adipokines [85]. In a similar study, an increase in IL-6 and TNF-α was associated with poor exercise practice and nutritional habits in people with sarcopenia [86]. Moreover, an increase in interleukin-10 (IL-10, anti-inflammatory cytokine) was observed, probably as a compensatory mechanism in sarcopenic people [87].

In this context, it has been noted that the systemic, chronic, subclinical, and low-grade inflammatory process associated with aging, known as inflammation, may be the most significant risk factor for the development and progression of the most common age-related diseases, frailty, and ultimately, death. On the other hand, adopting a healthy lifestyle can have an epigenetic impact and enhance anti-inflammatory and anti-fragility mechanisms [88].

### 5.2. Sarcopenic Obesity

Sarcopenic obesity (SO) occurs when an individual experiences both excessive body fat and low muscle mass and function. This combined condition increases the risk of functional impairment, development of metabolic diseases, and death compared to only one of these factors. The development of SO is linked to physical inactivity, insulin resistance, changes in hormone production, and overconsumption of energy [89,90]. One of the factors involved in the pathophysiology of SO is OxS, through an imbalance in the regulation of muscle mass (increased catabolism and decreased muscle anabolism), stress of the endoplasmic reticulum (which favors the overactivation of unfolded proteins, aggregation, and decrease in protein synthesis), as well as mitochondrial dysfunction (decreased ability to eliminate defective mitochondria and increased mtDNA damage). Similarly, OxS increases the activity of the ubiquitin-proteasome system (UPS) and activates caspases, impacting the regenerative function of satellite cells [91].

Elevated levels of fibrinogen and pro-inflammatory markers, such as IL-1, IL-6, CRP, and TNF-α are commonly observed in individuals with SO [92,93]. The cytokine IL-1 is involved in muscle catabolism via UPS, which increases the expression of atrogin-1 (an enzyme that is significantly expressed in muscle atrophy) and reduces myofibrillar proteins. On the other hand, low levels of IL-6 promote the turnover of muscle proteins, while high levels lead to skeletal muscle wasting. The increase in CRP contributes to a decrease in the proliferation rate of myoblasts. TNF-α induces transcriptional activation of nuclear factor-κB (NF-κB), generating ubiquitin-dependent muscle proteolysis. Overall, these cytokines contribute to muscle degradation by causing increased muscle catabolism, decreased muscle protein synthesis, and chronic systemic inflammation, leading to the loss of muscle mass and function [94,95,96,97,98,99,100].

## 6. Telomeres

Telomeres are heterochromatic structures located at the ends of the chromosomes, and are integrated by TTAGGG sequences in double-strand DNA tandem, with around 2 to 20 kb and 50 to 200 bases in the monocatenary outgoing strand. These sequences are protected by a specialized protein complex, called “shelterin”, which is comprised of six proteins that are assembled along the final portion of telomeres, known as follows: Telomeric repeat-binding factor 1 and 2 (TRF1 and TRF2), protection of telomeres protein (POT1), TRF1-interacting nuclear factor 2 (TIN2), adrenocortical dysplasia protein homolog (TPP1), and repressor activator protein 1 (RAP1) [101,102]. 

In addition to protection, another function of shelterin is maintenance, which is the prevention of DNA from recognition as a damaged DSB site by the kinases ataxia-telangiectasia mutated (ATM), as well as ataxia telangiectasia and rad3-related protein (ATR). Similarly, it recruits the enzyme telomerase [101,102,103]. 

TRF1 and TRF2 proteins are united directly to double-chain telomeric repetitions in the form of homodimers with the TRF homology subdomain (TRFH) [104]. TRF1 is associated with other proteins, such as TIN2, POT1-interacting protein 1 (PIP1), and POT1, and interacts with TIN2 in the nucleus and the chromosomes in the metaphase. In turn, TIN2 interacts with TRF2. Therefore, a defective association of this protein with the factors of binding to telomeric repeats induces a DNA damage response (DDR) and POT1, and interacts with TIN2 in the nucleus and the chromosomes in the metaphase, destabilizing its union in the telomeric region [105,106]. 

Another function of TRF1 is to recruit a protein necessary for the regulation of telomeres in human cells called POT1, by joining the outgoing single strand that is 3′-rich in guanines [107]. Another protein that acts negatively to regulate the length of telomeres and interacts with POT1 and TIN2 is the PIP1 protein. It has been observed that the inhibition of PIP1 and POT1 through infection with the short hairpin RNA (shRNA) viral vector leads to the elongation of telomeres [108,109].

On the other hand, TRF1 binds to the tankyrase protein (TNKS) and accepts the adenosine diphosphate (ADP)-ribosylation, which is catalyzed by the tankyrase-poly (ADP-ribose) polymerase complex (TANKs-PARP).

RAP1 telomeric in mammals is associated exclusively with TRF2. In addition, beyond its function for the maintenance of telomeres in the nucleus, it has various pleiotropic functions in different physiological and pathological conditions associated with metabolism, inflammation, and OxS, which is its main function to protect the ends of the chromosomes of degradation, as well as the unwanted DNA repairs and chromosomal mergers [110,111,112]. On the other hand, it has been observed that TRF2 levels are decreased as neonatal human fibroblasts grow naturally. Similarly, RAP1 decreases but to a lesser extent, which occurs only in the nuclear fraction of old fibroblasts subjected to OxS with de hydrogen peroxide (H_2_O_2_) and not in the cytoplasmic. This is due to the functions of RAP1, as a positive regulator of the kappa-right-chain-enhancer of activated B cells (NFҡB) [113,114].

In general, it has been observed that when acute OxS occurs in the telomeric region, there is a decrease in TRF1 and TRF2 that could be contributing to its erosion [115]. The function of distinguishing between the natural extremes of the chromosomes with the DSB is regulated mainly by TRF2 and POT1, which prevents the cascades from damage signaling to the repair means of DNA and DSB [116]. T loop formation by TRF2 kidnaps the 3′-end of the chromosomes, which prevents the recognition of the DDR [117]. In addition, if the DDR is activated, it causes the arrest of the cell cycle until the damage is repaired and if the response persists, it can induce senescence [118].

### 6.1. Telomerase

The telomerase was biochemically identified by Greider and Blackburn, who showed that the synthesis of the enzyme was based on an RNA that serves as a guide for the polymerization of telomeric DNA sequences [119,120]. This is attributed to the elongation of the telomeres with the help of the TRF1 protein, which recruits TIN2 and acts as the damping mechanism for the elongation of the same protein [105,106]. This occurs through ribosylation-ADP (a post-translational modification that involves the addition of one or more ADP-ribose) of TRF1, reducing its ability to join the telomeric DNA, and in turn, the above allows for telomerase to extend telomeres and life is extended [121,122,123].

Telomerase is a polymerase DNA that extends the 3′-end of the chromosome by synthesis of multiple telomeric repetitions. Moreover, it is the only ribonucleoprotein that contains RNA reverse transcriptase (TERT) and human telomerase RNA component (TERC) [124]. Ribonucleoprotein TERT is active in progenitor and carcinogenic cells, although it is also located in somatic cells, where its activity is very low or null [125]. In addition, telomerase is constituted by some associated proteins for proper functioning, such as dyskerin (DKC1), telomerase Cajal body protein (TCAB1), and ribonucleoproteins: H/ACA ribonucleoprotein complex subunit 3 (NOP10), H/ACA ribonucleoprotein complex subunit 2 (NHP2), and H/ACA ribonucleoprotein complex subunit 1 (GAR1).

The loss of function of these proteins has been associated with different patterns of inheritance, such as ribosomal disease (impaired ribosome biogenesis and function), including congenital dyskeratosis (caused by poor maintenance of the telomeric length resulting in short telomeres), cartilage hair hypoplasia-anauxetic dysplasia (characterized by short limbs, sparse hypoplastic hair, defective T-cell immunity, hypoplastic anemia, and an increased risk in developing malignancies) [126,127]. The silencing of DKC1 and NOP10 decreases the activity of telomerase with the increased GSSG/GSH ratio, protein carbonylation, and high expression of manganese SOD (MnSOD), suggesting that the loss of the functions of DKC1 and NOP10 induces OxS independently to the telomere shortening [128] (Figure 2). Similarly, DKC1 mutations increase OxS and DDR [129]. There is scientific evidence, which sustains that during each cell division and due to the low expression of the telomerase enzyme, telomeres are gradually shortened [130,131]. It has been determined that during each replication round, telomeres are shortened on average between 30 and 200 base pairs, where only 10 base pairs are lost due to the problem of final replication, while the rest is as a consequence of OxS in somatic cells [132,133]. In addition, the age-dependent telomere shortening can be decelerated by suppressing intracellular OxS or by DNA repair mechanisms [134].

### 6.2. Sarcopenia and Telomeric Length

There is a causal relationship between telomeric length, loss of skeletal muscle mass, frailty, and OxS in older adults with multiple morbidities and sarcopenia [57,135]. However, it has been observed that shorter telomeric lengths are associated with cardiovascular disorders (blood pressure, aging, and coronary arteries disease), T2DM, OxS damage, and inflammation [136]. In an observational study carried out over 12 months, OxS markers, total antioxidant capacity, telomeric length, and apoptosis in peripheral blood samples of poly-pathological patients (heart disease, autoimmune, pulmonary, neurological, etc.) were evaluated with frailty and/or sarcopenia with an average age of 77.3 years, where it was observed that there was a significant decrease in total antioxidant capacity and telomeric length, with an increase in SOD, without evidence of apoptosis compared to non-sarcopenic or fragile patients [135] (Table 1).

In a prospective study (follow-up for 5 years) conducted in older adults, no association between the telomeric length and the diagnosis of sarcopenia was observed; however, longer telomeric lengths are associated with greater grip strength [137]. Moreover, it has been observed that in mononuclear cells of the peripheral blood of people with sarcopenia, the telomeric length is shorter compared to non-sarcopenic people [135]. On the other hand, middle-aged people (44.7 years) with sarcopenic obesity expressed high adiposity, low muscle mass, and significantly shorter telomeres, as compared to the reference group [140]. Molecular studies indicate that the TERRA expression (long non-coding RNA consisting of UUAGGG repetitions and telomerase regulator) was lower in sarcopenic participants compared to non-sarcopenic, while the exercise-nutrition intervention increased the TERRA expression, although this did not significantly increase the length of telomeres in sarcopenic people [138,141]. On the other hand, in experimental models, it has been shown that the absence of an enzyme that reduces peroxides with a preserved cysteine residue and protects OxS cells called peroxirredoxin-6 (PRDX6) (Prdx6−/− mice), induced T2DM, dramatically reduced the telomeric length with increased activity of senescence-associated beta-galactosidase (SA-β-gal) (biomarker for aging and senescent cells), and decreased nuclear-cytoplasmic transport of SIRT1 (important enzyme for the replacement of defective mitochondria). Moreover, it has been observed that the absence of PRDX6 results in decreased grip strength and reduction in the cross-sectional area of muscle fibers compared to the control group. However, in this study, sarcopenia may be due to the consequence of the T2DM and not to the normal aging process, which would indicate metabolic sarcopenia and may be associated with the increase in OxS with the loss of PRDX6 [142,143,144,145,146,147,148,149].

In regard to the inflammatory process, it has been observed that an increase in TNF-α, IL-1β, IL-6, and CRP can contribute to telomeric attrition [76,77,78,79] through an inflammatory waterfall activated by NF-κB, which in turn, regulates telomerase expression, as well as the protein belonging to the shelterin RAP1 complex [114].

## 7. Base Excision Repair (BER)

It is estimated that the number of injuries suffered by a normal cell per day is 1000 molecules of 8-OHdG or 8-oxodG, and the base excision repair (BER) is one of the most important roads involved in correcting oxidative lesions to the DNA (Table 2) [150,151]. This repair route is subdivided into two pathways: The short patch pathway and the long patch pathway, which differ in the number of nucleotides that are split and the number of enzymes that participate in them. On the short patch pathway, only a nucleotide is removed and incorporated, while in the long patch pathway, from two to six nucleotides are incorporated. The short route begins when the damaged DNA is recognized by a DNA glycosylase, which splits the N-glycosyl link that joins the nitrogenous base with the backbone of the deoxyribose phosphate. This split generates an abasic/apurinic/apyrimidinic or AP site, which is processed by an AP endonuclease. The AP endonuclease cuts the phosphodiester bond at the 5‘-end of the AP site, thus generating an SSB, which has a free hydroxyl at the 3′-end and a phosphate group at the 5′-end to deoxyribose sugar [152]. It is a requirement that the SSB has the 3′-hydroxyl and 5′-phosphate ends, in order that the polymerase DNA incorporates the corresponding nucleotide, and thus the link can seal the loose ends of the DNA. The long patch pathway requires a greater amount of enzymes to finish the repair, such as the proliferating cell nuclear antigen (PCNA), which serves as a scaffolding protein for polymerase, the DNA ligase, and the exonuclease FEN 1. The latter was in charge of removing the strand that was hanging when the DNA polymerase added the nucleotides to the broken strand. It has been observed that cells which lack the enzyme that eliminates the adduct 8-oxodG (OGG1) increase cell fragility without compromising its survival [153].

### 7.1. Nucleotide Excision Repair (NER)

NER is an essential method to repair bulky bases damaged in the DNA (Table 2), such as adducts 4-(methylnitrosamino)-1-(3-pyridyl)-1-butanone (NNK) and N-nitrosonornicotine (NNN) [154]. In this route, the steps for reparation are as follows: (i) Recognition of the damage; (ii) double incision (3‘ and 5′) of the damaged monocatenary fragment (24–32 nucleotides); (iii) release of the damaged oligomer; (iv) synthesis to fill the hole; and (v) sealing of the loose ends of the DNA. NER is a repair system capable of eliminating injuries that distort DNA, such as UV-induced photoproducts, cyclobutane pyrimidine dimers, as well as pyrimidine photoproduct 6–4 pyrimidine and DNA adducts induced by chemicals as aflatoxins [155,156,157].

NER is also subdivided into two pathways: (1) The repair of the global genome (GG), which repairs the lesions throughout the genome; and (2) the traffic coupled (TC), which is limited to eliminating injuries that block transcription [158].

The GG repair begins with the recognition of the damage, which occurs with the protein complex CUL4-DDB (XPE), consisting of the DNA damage-binding protein 1 (DDB1). This protein participates in the union to damage DNA and is associated with DDB2, which is a complex that can recognize the damage induced by UV, preferably to cyclobutane pyrimidine dimers. Cullin 4 (Cul4) is a member of the ubiquitin ligase E family, in which ubiquitin to the H3 and H4 histones weakens the histones-DNA union and facilitates the recruitment of repair proteins [159,160,161].

The GG can also recognize injuries through the complex subunit, DNA damage recognition and repair factor (XPC), homolog B, nucleotide excision repair protein (HR23B), and centrin-2 (CETN2). Moreover, XPC interacts with XPE to complete the damage recognition step, as well as with the basal transcription factor TFIIH [160]. This factor is responsible for opening the DNA that surrounds the damaged base, with an helicase activity. TFIIH is composed of cyclin-dependent kinase 7 (CDK7), DNA excision repair protein ERCC-3 (XPB), transcription initiation factor TFIIH subunit 1 (TFIIH1), CDK-activating kinase assembly factor MAT1 (MNAT1), DNA excision repair protein ERCC-2 (XPD), transcription initiation factor TFIIH subunit 2 (TFIIH2), cyclin H (CCNH), TFIIH basal transcription factor complex TTD-A subunit (TTDA), transcription initiation factor TFIIH subunit 3 (TFIIH3), and transcription initiation factor TFIIH subunit 4 (TFIIH4). At a later time, this gives way to the incision through the nucleases DNA excision repair protein ERCC-4 (XPF) and DNA excision repair protein (ERCC-1), and then, the PCNA and replication factor C subunit 1 (RFC) are split. Finally, the synthesis of DNA polymerase (Polδ and Polε) and union is given by Lig1 [160,162].

### 7.2. Telomere Repair Mechanisms

OxS damage to the DNA with the formation of 8-oxodG and 8-OHdG molecules represents the most frequent damage to human cells, especially at a telomeric level [115]. Moreover, the erosion of telomeres is associated with the presence of infectious diseases and the mechanisms that protect the organism, since the inflammatory process can induce OxS [76].

In vitro studies in U2OS cells point out that when DNA damage in telomeres is not repaired efficiently, compared to the DNA damage in non-telomeric regions, its length is shortened. In HeLa cells, the production of oxidative damage, specifically in the telomeric region, induced by the pLVX-IRES-Puro KR-TRF1 vector, leads to cellular senescence or death. The specific damage of telomeres induces chromosomal aberrations, including the loss of chromatid telomers. In general, OxS damage induces the dysfunction of telomeres and underlines the importance of maintaining the integrity of telomeres against this type of damage [163]. Concerning the BER repair mechanism, it has been documented that in cells that lack OGG1, the chronic 8-oxodG formation increases the fragility and shortening of telomeres, altering cell growth [153].

The NEIL3 glycosylase protects the stability of the genome by the directed repair of oxidative damage in telomeres during the S/G2 phase. It has been observed that NEIL3 is colocalized with TRF2 during phase S and this association increases with OxS. Moreover, its recruitment can be through TRF1 and its interaction improves its activity [164]. Furthermore, it has been proven that the telomeric DNA containing thymine glycol, 8-oxodG, or spiriminodihydantoin can generate quadruplex DNA in the telomeric region and only NEIL3 can split injuries with thymine glycol in this type of DNA. Therefore, NEIL1 shows better activity in the repair of DNA injuries with guanidinohydantoin [165].

On the other hand, the XPF-ERCC1 endonuclease that participates in the NER pathway, in the repair of cross bonds, and in homologous recombination, can interact with TRF2 and function as a negative regulator in the maintenance of telomeres length [166]. 

It has been observed that the expression of TRF2 in mice skin (K5-TRF2) has a severe phenotype, such as premature skin deterioration, hyperpigmentation, and increased cancer, which is similar to xeroderma pigmentosum syndrome in humans. Similarly, the skin cells of these mice report a marked shortening of telomeres and increased chromosomal instability compared to wild-type mice (WT). On the other hand, the strain of mutant mice ERCC4−/− (a gene that encodes for XPF) has the same telomeric length compared to the WT strain and K5-TRF2 ERCC+/+ mice. However, they have a marked reduction in the telomeric length. This demonstrates the link between telomeres and damage repair, in which this alteration is the basis of genomic instability, cancer, and aging [167].

## 8. Sarcopenia Prevention Interventions

To date, a causal relationship between OxS and the aging process has not been fully established; however, this does not indicate that the reduction in oxidative damage is not intended as an alternative to promote a healthy life, delaying sarcopenia, and its associated events [168]. Similarly, various therapies have been implemented to counteract the effects of sarcopenia through the administration of food supplements (mainly proteins), antioxidants, anti-inflammatories, physical activity, resistance training, and caloric restriction [79,85]. It has been shown that supplementation with resveratrol for 10 months at 0.05% increases the enzymatic activity of MnSOD, and reduces the levels of lipoperoxidation and H_2_O_2_, without significant changes in carbonylated proteins in muscle samples, suggesting a protective effect against OxS as evidenced by the correct functioning of fast twitch fibers. Of note, supplementation did not improve strength or reduce muscle loss [169]. Another therapy with promising results is growth hormone (GH) replacement, which reduced oxidative damage at the DNA level by decreasing the levels of 8-OHdG and carbonylated proteins in skeletal muscle [170].

In humans, interventions with high-quality protein supplements, such as whey that has a large amount of essential amino acids including leucine, as well as resistance training, are promising treatments to counteract sarcopenia [171,172]. Nutritional intervention for 13 weeks with whey protein (20 g) supplemented with vitamin D (800 IU), total leucine (3 g), carbohydrates (9 g), and fat (3 g) improved lower extremity function and muscle mass in sarcopenic patients [173], although it was subsequently shown that the necessary basal levels of vitamin D and proteins are required, in order that the aforementioned is possible [174]. Another vitamin that improves muscle strength in sarcopenic older adults is vitamin E, coupled with a decrease in proinflammatory markers, such as interleukin-2 (IL-2) and insulin-like growth factor 1 (IGF-1) [175]. Similarly, in obese sarcopenic women, it was observed that caloric restriction and supplementation with whey protein and leucine improve strength and muscle mass [176]. Nutraceuticals are products derived from human food that claim to provide health benefits, but there is insufficient evidence to support these claims [177]. For example, taking vitamin D supplements is generally considered as safe, since toxicity from high levels of vitamin D in the blood is rare. However, long-term use of vitamin D can result in hypercalcemia, hypercalciuria, and hyperphosphatemia, which are indicators of vitamin D overdose. [178]. Supplementation with collagen peptides (15 g/day) combined with resistance training (three sessions per week) for 3 months improves muscle mass and strength in sarcopenic individuals [179]. Moreover, the leucine metabolite β-hydroxy-β-methylbutyrate (HMB) (3 g/day) has been shown to preserve muscle mass in healthy older adults after only 10 days of supplementation [180]. However, a systematic review found that HMB supplementation has no impact on muscle strength or body composition in adults aged 50 to 80 years [181].

It should be noted that there are several promising pharmacotherapies for countering sarcopenia, but combining these therapies with exercise, nutritional supplements, and a balanced diet enhances their effects. One therapy targeting the myostatin-ActRII pathway is the intravenous administration of bimagrumab (700 mg per month) for 6 months, which moderately increases lean body mass by 7%, with no apparent changes in physical performance or gait speed, as compared to the placebo group receiving a balanced diet and light exercise [182,183]. Another promising monoclonal antibody is LY2495655, which has a beneficial effect on lean mass and physical performance [184]. The consumption of the androgen receptor modulator MK-0773 (50 mg two times per day for 6 months) shows an increase in lean body mass compared to the placebo, but with no changes in muscle strength or function [185].

Similarly, in sarcopenic experimental models, administration of apelin has been shown to improve mitochondrial biogenesis and autophagy in myofibers, as well as regeneration of satellite cells, through stimulation of the trophic forkhead box O3 (FOXO3)-MURF-1-atrogin axis and activation of AMP-activated kinase (AMPK) [186]. Additionally, a combination of HMB supplementation and low-magnitude high-frequency vibration (LMHFV) for 3 months has been shown to enhance muscle strength and suppress the Wnt/β-catenin signaling pathway that promotes adipogenesis [187].

Caloric restriction (CR) without malnutrition refers to a reduction in energy intake between 20% and 40% [188,189]. CR favors hormesis, delaying the appearance of age-related diseases, including sarcopenia, and prolonging lifespan. By reducing glucocorticoid-mediated inflammation, it increases serum cortisol (with anti-inflammatory activity) and decreases TNF-α transcription (proinflammatory cytokine), inhibits apoptosis by reducing caspase-3 and caspase-8 (enzymes that induce programmed cell death), as well as activates autophagy by increasing the expression of autophagic markers beclin-1 and microtubule-associated protein 1A/1B-light chain 3 (LC3) [190,191,192]. Moreover, it decreases FR levels at the mitochondrial level and oxidative damage at the mtDNA level [193]. Furthermore, it suppresses muscle atrophy in both slow and fast muscle fibers of the soleus muscle [194], which leads to the improvement in quality and cell function in skeletal muscle and a delay in frailty.

### 8.1. Physical Exercise

Exercise, unlike physical activity, is a subset of planned, structured, and repetitive physical activity that aims to improve or maintain physical fitness. Physical activity is any bodily movement produced by skeletal muscles that result in energy expenditure, and includes activities of daily life that are carried out at home, transportation, and recreational or occupational [195]. The American College of Sports Medicine (ACSM) recommends moderate- to vigorous-intensity cardiorespiratory exercise for adults to achieve a minimum total energy expenditure of 500–1000 MET/min/week. This can be achieved through moderate-intensity exercise for at least 30 min a day, on at least 5 days a week (totaling at least 150 min a week) or vigorous-intensity exercise for at least 20 min a day, on at least 3 days a week (totaling at least 75 min a week), or a combination of both [196]. To maintain strength, flexibility, and muscular endurance, training programs that include aerobic, resistance, and flexibility training are recommended. In this context, physical activity is one of the most important interventions to treat sarcopenia, since it assists in improving muscle structure and function, preventing disability and frailty, with beneficial effects on metabolic and cardiovascular diseases in the elderly [197]. It has been suggested that regular physical activity may be essential in maintaining the anabolic response to protein intake with aging [80]. 

### 8.2. Exercise and Sarcopenia

There is sufficient scientific evidence that highlights the beneficial effect of physical activity on skeletal muscle health. In addition, it is pointed out that unless you have an active lifestyle, the mitochondrial decline in muscle cells will occur as you age [63]. Therefore, exercise has been used as a strategy to prevent or delay sarcopenia. On the one hand, it seems to reduce the loss of muscle mass. On the other hand, it increases the aerobic metabolism, and thus promotes mechanical damage due to free radicals accumulation [38]. To avoid the risk of falls, balance, strength, and resistance training are recommended to facilitate muscle protein anabolism [198]. Another strategy to prevent loss of muscle mass is to avoid losing weight, since the quantities and quality of proteins are of paramount importance for the correct synthesis of muscle proteins. Moreover, it has been suggested that the consumption of antioxidants, vitamin D, and fatty acids may contribute to the maintenance of muscle function [199].

### 8.3. Exercise and Telomere Length

Studies carried out on lifestyles based on nutritional recommendations, physical training programs (aerobic, strength, and balance), cognitive training, and management of metabolic risk factors allow for the maintenance of the telomeric length in the peripheral blood of older adults (Figure 3). In addition to nutritional and exercise interventions, telomeric length increased in overweight and obese children and adolescents [200,201]. Although it appears that exercise can induce apparent telomere lengthening, the mechanisms by which this occurs remain unknown [202]. It is well known that chromosomal aberrations and instability are more frequent in the older adult population, which can lead to cell death, releasing intracellular content. Among them, genomic DNA can be found to be circulating in the bloodstream (cfDNA) and integrated into the genome of the cells of the organism, thus increasing genomic instability [203]. Exercise can lead to a rapid increase in cfDNA, which is released mainly or almost exclusively by granulocytes with overtraining or strenuous training, returning to its basal levels almost immediately [204,205]. In a study on the telomeric length and its possible association with cfDNA, they did not find any type of correlation between them; however, there was an association between improved chair lift testing with longer telomeric lengths in people who underwent resistance training with high-protein supplements [206]. Another study showed that longer telomeres are positively associated with greater physical activity and the most active subjects had 200 more nucleotides in the telomeric region than the least active individuals [207]. A randomized controlled trial showed that 1 year of aerobic exercise did not cause changes in telomeric length among the exercise group compared to the control. Similarly, diet and exercise-based weight loss did not change telomeric length telomeres in both studies conducted on postmenopausal women [208,209]. Although the previously mentioned studies are based on the telomeric region of blood cells, not muscle cells, it has been evidenced that the length of telomeres of leukocytes and skeletal muscle cells can be positively related to a healthy life and inversely correlated with a greater risk of various age-related diseases [5].

### 8.4. Exercise and DNA Repair

There is clear evidence that an acute period of intense exercise generates enough ROS to challenge the body’s antioxidant defense system. In general, activation of redox-sensitive pathways results in gene products that restore intracellular oxidant-antioxidant homeostasis, such as the expression of genes that code for SOD and GPx [210]. It has been observed that aerobic exercise can cause DNA damage in athletes associated with the intensity and the distance covered; namely, DNA lesions were more evident when running a distance of 42 km than a distance of 5 and 10 km [211]. A study found that 12 weeks of moderate-intensity, low-frequency explosive-type resistance training had significant benefits for older people (70–75 years), improving their muscle strength and power compared to the control group (without exercise). Additionally, the exercise group had a higher proportion of reduced glutathione, which improved the redox state, with low levels of malondialdehyde and protein carbonylation [212]. Physical exercise has many health benefits; however, its properties will depend on the type and intensity of the exercise routines. In this context, various studies indicate how DNA repair mechanisms are activated by physical activity (Table 3).

## 9. Conclusions

Physical exercise is a crucial intervention for delaying the loss of skeletal muscle mass. It stimulates the growth of muscle fibers, prevents the conversion of fast fibers into slow fibers, and reduces the decline in satellite cells. Moreover, physical activity generates the hormetic state, which promotes pro-oxidation and stimulates the production of antioxidant enzymes. Studies have shown that physical training has protective effects on the DNA of lymphocytes, which may be related to an increase in antioxidant capacity and a decrease in DNA strand breaks and formamidopyrimidine DNA glycosylase (FPG) sensitive sites. It has been suggested that lesions caused by guanine oxidation in telomeric DNA can be eliminated by the glycosylases NEIL1 and NEIL3, since OGG glycosylase activity has been reported to be null in this region. It is clear that physical exercise prevents the wear and tear of telomeres and, in some cases, lengthens them. However, the effect of exercise on DNA repair in the telomeric region and its connection to sarcopenia is not yet fully understood. Therefore, further translational research is needed to better understand these mechanisms.

## Figures and Tables

**Figure 1 biomedicines-11-00598-f001:**
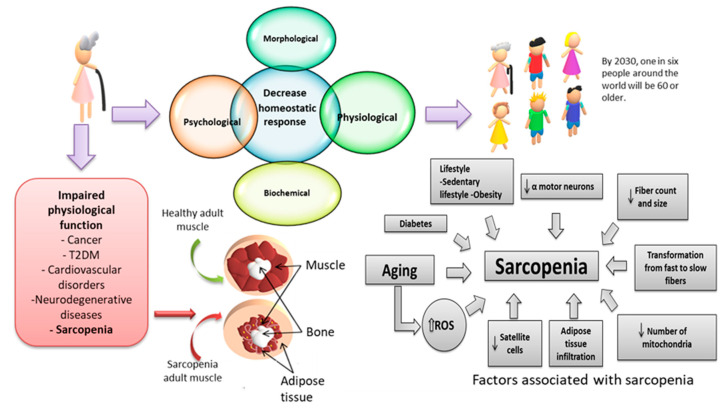
Muscular changes associated with aging. Muscle mass declines approximately 3–8% per decade from the age of 30, whose alteration is related to an increase in reactive oxygen species (ROS) and inflammation linked to a sedentary lifestyle [45]. T2DM: Type 2 diabetes mellitus.

**Figure 2 biomedicines-11-00598-f002:**
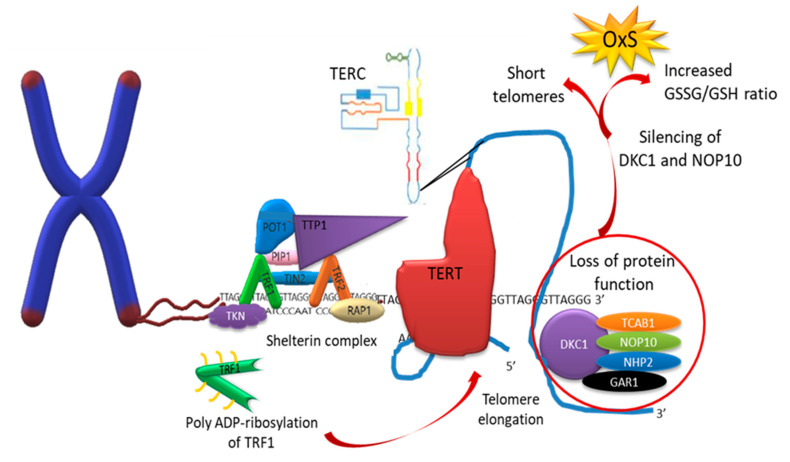
Representation of the telomerase. Telomerase is a DNA polymerase responsible for extending the 3′-end of chromosomes. It is composed of TERT reverse transcriptase, TERC telomeric RNA, and associated proteins, including DCK1, NOP10, NHP2, and GAR1. The poly ADP-ribosylation of TRF1 stimulates telomere elongation, whereas a loss of function of the associated proteins can lead to telomere shortening. Silencing of DCK1 and NOP10 can also cause oxidative stress (OxS). Key components of sheltering and telomerase complex are described as follows: TRF1 (telomeric repeat-binding factor 1); TRF2 (telomeric repeat-binding factor 2); POT1 (protection of telomeres protein); PIP1 (POT1-interacting protein 1); TIN2 (TRF1-interacting nuclear factor 2); RAP1 (repressor activator protein 1); TERC (human telomerase RNA component); TCAB1 (telomerase Cajal body protein); NOP10 (H/ACA ribonucleoprotein complex subunit 3); NHP2 (H/ACA ribonucleoprotein complex subunit 2); GAR1 (H/ACA ribonucleoprotein complex subunit 1); DCK1 (dyskerin); and TNKS (tankyrase protein).

**Figure 3 biomedicines-11-00598-f003:**
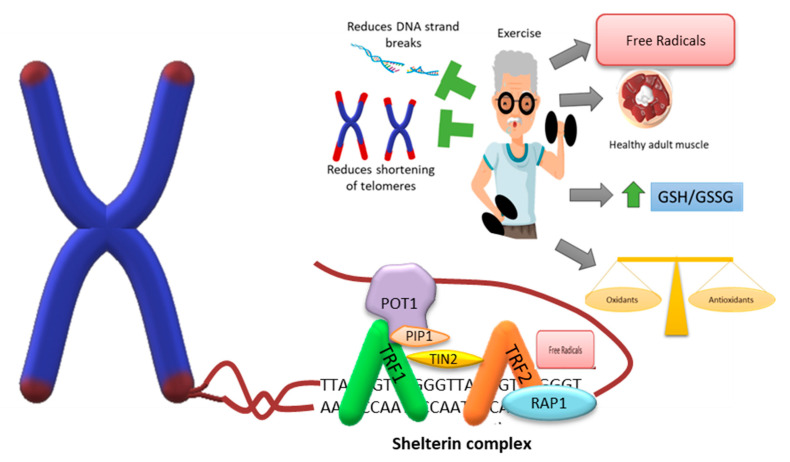
Representation of the telomere-shelterin association and effect of exercise. The ends of the chromosomes are protected by the shelterin protein complex, which prevents them from being recognized as damaged sites and from being erroneously repaired and recognized as double-strand breaks, and can generate chromosome instability. Exercise assist in maintaining or improving physical activity and is one of the main interventions for sarcopenia. Its effect lies in preventing the decrease in satellite cells, the increase in fatty tissue infiltrates in muscle fibers, and assists in obtaining bone mass, muscle mass, and skeletal muscle strength. Although exercise generates an increase in the formation of free radicals, it maintains the balance between the number of pro-oxidants and antioxidants. It also helps maintain and increase the length of telomeres, thus preventing erosion of the ends of chromosomes. Key components of the sheltering: TRF1 (telomeric repeat-binding factor 1); TRF2 (telomeric repeat-binding factor 2); POT1 (protection of telomeres protein); PIP1 (POT1-interacting protein 1); TIN2 (TRF1-interacting nuclear factor 2); and RAP1 (repressor activator protein 1).

**Table 1 biomedicines-11-00598-t001:** Association of telomeric length and sarcopenia.

Population Sarcopenia	Determinations	Objective	Findings	Ref.
142 persons aged ≥65 years.	The presence of sarcopenia was established according to the EWGSOP. Whole-body fat-free mass was measured by BIA. The frailty status of participants was assessed according to both Fried’s criteria and the elative telomere length of qRT-PCR.	Determine whether PBMC telomeres obtained from sarcopenic older persons were shorter relative to non-sarcopenic peers.	PBMC telomere length, expressed as T/S values, is shorter in older outpatients with sarcopenia. The cross-sectional assessment of PBMC telomere length is not sufficient for capturing the complex, multidimensional syndrome of frailty.	[135]
The stratified sample includes a total of 976 males and 1030 females, in order that approximately 33% would each be aged 65–69, 70–74, and 75 years and older.	Diagnosis of sarcopenia. The T/S ratio was assessed by qRT-PCR.	To examine the association between telomeric length and diagnosis of sarcopenia based on an appendicular skeletal mass index (ASMI), grip strength, walking speed, and chair sit-to-stand in a 5-year prospective study.	Longer telomere length was associated with a slower decline in grip strength in Chinese older persons.	[137]
36 sarcopenic people and 36 healthy people. Older adults (age ≥ 65 years).	Anthropometric measurement. Grip strength. Measurement of telomere length analysis was performed by qPCR. RNA isolation and quantification of TERRA.	To explore the impact of sarcopenia on telomere length and TERRA expression, and changes following exercise and nutrition intervention in the sarcopenic population.	No significant difference in telomere length between control subjects and participants with sarcopenia.	[138]
Included 444 patients with an average age of 77.3 ± 8.4 years.	Determination of sarcopenia. Determination of frailty by meeting three or more of Fried’s criteria. OxS markers. Telomere length. DNA fragmentation.	To explore the main markers of OxS, telomere length, and apoptosis parameters in a multicenter cohort of patients with multimorbidity in a hospital.	OxS markers and telomere length were enhanced and shortened, respectively, in blood samples of poly-pathological patients with sarcopenia and/or frailty. Both were associated with decreased survival.	[57]
20,400 older adults (average age: 67.79 ± 4.9 years, 53% male).	Baseline leukocyte telomere length was measured using a multiplex qPCR technique and expressed as a T/S ratio.	Examined the association between leukocyte telomere length and osteosarcopenia.	Telomere length was not associated with osteosarcopenia; however, a slow walking pace was associated.	[139]
5397 individuals; (average age: 44.7 years, 51.3% male).	Body composition evaluation using dual-energy X-ray absorptiometry (DXA). Evaluation of whole blood telomeric length by qPCR. Average telomere length is expressed as the ratio T/S.	Examine the relationship between sarcopenic obesity (SO) and telomere length (TL) in a representative adult population.	Sarcopenia and obesity may act synergistically to shorten telomeres.	[140]

EWGSOP: European working group on sarcopenia in older people; BIA: Bioelectrical impedance analysis; qRT-PCR: Quantitative real-time polymerase chain reaction; PBMC: Peripheral blood mononuclear cells; T/S: Telomere length/single copy gene; ASMI: Appendiceal skeletal mass index; TERRA: Telomeric repeat-containing RNA; OxS: Oxidative stress.

**Table 2 biomedicines-11-00598-t002:** Repair mechanisms of BER and NER.

Causing Damage			
ROS, alkylating agents, and UV		UV and ROS	
Injuries			
8-OHdG, 8-oxodG, alkylated bases, or mismatch bases		Pyrimidine dimers bulky lesions	
BER		NER	
Recognition of the damage		Recognition of the damage	
Short patch pathway	Long patch pathway	GG	TC
Bifunctional glycosylases OGG1, NTH, NEIL1, NEIL2, and NEIL3	Monofunctional glycosylases UNG, MUTY, MBD4, MPG, and SMUG	Complex Cul4-DDB (XPE): RBX1, Cul4, DDB1, DDB2 Complex XPC: XPC, HR23B, CETN2	Complex Cul4-CSA: RBX1, Cul4, DDB1, CSA, CSB
Chain excision sugar removal		Relaxed DNA	
APE1 and APE2	APE1 and APE2	Complex TFIIH: CDK7, XPB, TFIIH1, MNAT1, XPD, TFIIH2, CCNH, TTDA, TFIIH3, TFIIH4,	XPG, XPA, and RPA
Synthesis		Incision, excision, and synthesis of DNA Incision: XPF and ERCC1 Excision: PCNA, RFC Synthesis: Polδ and Polε	
Dpol, XRCC1, Lig3	Dpol, PCNA, Polδ, Polβ, Polε, Fen1		
Ligation Lig1			

ROS: Reactive oxygen species; BER: OGG1: 8-oxoguanina DNA glicosilasa; NTH: Nth-like DNA glycosylase 1; NEIL1: Nei-like DNA glycosylase 1; NEIL2: Nei-like DNA glycosylase 2; NEIL3: Nei-like DNA glycosylase 3; UNG: Uracil DNA glycosylase; MUTY: MutY DNA glycosylase; MBD4: Methyl-CpG binding domain 4 DNA glycosylase; MPG: N-methylpurine DNA glycosylase; SMUG: Single-strand-selective monofunctional uracil-DNA glycosylase 1; APE 1 and 2: Apurinic/apyrimidinic endonuclease 1 and 2; Dpol: Polimerase DNA; XRCC1: X-ray repair cross complementing 1; Lig3: DNA ligase 3; PCNA: Proliferating cell nuclear antigen; Polδ, Polβ, Polε: DNA polimerase; FEN1: Flap endonuclease-1; NER: RBX1: Ubiquitin-protein ligase RBX1; CUL4: Cullin 4; DDB1: DNA damage-binding protein 1; DDB2: DNA damage-binding protein 2; XPC: Complex subunit, DNA damage recognition and repair factor; HR23B: Homolog B, nucleotide excision repair protein; CETN2: Centrin-2; RBX1: E3 ubiquitin-protein ligase RBX1; CSA: DNA excision repair protein ERCC-8; CSB: DNA excision repair protein ERCC-6; CDK7: Cyclin-dependent kinase 7; XPB: DNA excision repair protein ERCC-3; TFII1–4: Transcription initiation factor TFIIH subunit 1–4; MNAT1: CDK-activating kinase assembly factor MAT1; CCNH: Cyclin H; TTDA: TFIIH basal transcription factor complex TTD-A subunit; XPG: DNA excision repair protein ERCC-5; XPA: DNA-repair protein complementing XPA cells; RPA: Replication factor A1; XPF: DNA excision repair protein ERCC-4; ERCC1: DNA excision repair protein ERCC-1; RFC: Replication factor C subunit 1. Database Kyoto Encyclopedia of Genes and Genomes (KEEG).

**Table 3 biomedicines-11-00598-t003:** Physical exercise and DNA repair.

Population	Determinations	Objective	Finding	Ref.
Fifty-seven healthy males (40 to 74 years)	Strength tests. Power tests. DNA damage. Assessment of repair capacity. Lipid peroxidation. TAC.	This study aimed to determine the effects of a 16-week combined physical training program on DNA damage and DNA repair of human lymphocytes, taking into account the improvement of physical fitness. To investigate the role of OxS in these changes.	Improvement in general physical performance in the experimental group. Decrease in DNA chain breaks and sites, sensitive to formamide-pyrimidine glycosylase, with a concomitant increase in antioxidant activity and a decrease in lipid peroxidation levels after physical training. There are no significant changes in the enzymatic activity of DNA glycosylase and 8-oxoguanine.	[213]
Endurance-trained and young healthy males (age 20 to 36 years)	Simple DNA single break detection. Poly detection (ADP-ribose) and phosphorylation of the H2AX histone (γh2ax).	Determine the general effect of acute exhaustive exercise and physical aptitude (aerobic capacity) on DNA damage, radiosensitivity, and PLP1 activity induced by radiation in immune cells isolated from trained and non-healthy trained volunteers.	Acute exercise induces DNA strand breaks in lymphocytes in untrained individuals. During acute exercise, trained subjects repaired radiation-induced DNA strand breaks more rapidly than untrained subjects. Trained subjects maintained higher levels of radiation-induced PARP1 activity after acute stress.	[214]
Thirty-two healthy Caucasian males (40 to 74 years)	Assessment of strand break DNA (SB) and oxidative damage to DNA. Evaluation of sites sensitive to FPG. Assessment of repair capacity with the comet assay. The activity of OGG1. TAC. Determination of the hOGG1 (Ser326Cys) polymorphism.	To investigate the possible influence of genetic polymorphisms of hOGG1 on DNA damage and repair activity OGG1 enzyme in response to 16 weeks of combined physical training.	At baseline, there were no differences in DNA damage and OGG1 activity between the groups. With 16 weeks of physical exercise, there was a decrease in DNA strand breaks in both groups, as well as a decrease in FPG-sensitive sites and an increase in TAC in WTG.	[215]
Fourteen (apparently healthy recreationally active males (age 22 ± 2 years, stature 178 ± 6 cm, mass 83 ± 8 kg, BMI 26.2 ± 2)	DNA single-strand breaks and FPG-sensitive sites. Detection of double-strand breaks via histone γ-H2AX and 53BP1. Lipid hydroperoxides. Soluble antioxidants. EPR.	Characterization of the interplay of exercise and hypoxia about DNA damage repair. Quantification of the effects of exercise in hypoxia on single- and double-strand DNA damage using the comet assay in conjunction with γ-H2AX and colocalized repair protein 53BP1.	Increase in γ-H2AX and 53BP1 foci after high-intensity exercise, with markers, increased in hypoxia. Although normoxia resulted in a marked increase in foci detection, hypoxia challenge resulted in a 2.5- and 3.5-fold increase in γ-H2AX and 53BP1 foci, respectively, after exercise.	[216]
Sixty-one T2DM subjects, aged (mean ± SD: 50.3 ± 4.2)	Glycemic status. DNA damage (Comet assay). Oxidative DNA damage. OGG1 protein expression. TAC.	Elucidation of the mechanism of action of yoga on T2DM-related DNA damage in terms of its effect on oxidative DNA damage and DNA repair markers.	The yoga group showed a significant reduction in DNA damage, oxidative DNA damage marker, and fasting blood sugar compared to the control. The beneficial effect of yoga on DNA damage in T2DM subjects was found to be mediated by the mitigation of oxidative DNA damage and enhancement of DNA repair.	[217]

TAC: Total antioxidant capacity; OxS: Oxidative stress; γH2AX: H2AX histone; PLP1: Polymerase poly; PARP: Poly(ADP-ribose) polymerase; FPG: Formamidopyrimidine DNA glycosylase; OGG1: 8-Oxoguanine DNA glycosylase; WTG: Wild-type group; BMI: Body mass index; 53BP1: 53-Binding protein 1; EPR: Electron paramagnetic resonance spectroscopy; T2DM: Diabetes mellitus type 2.

## Data Availability

Not applicable.

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
