# Peer review of "Aging, Physical Exercise, Telomeres, and Sarcopenia: A Narrative Review"

_biomedicines, 2023, doi:10.3390/biomedicines11020598_

Round 1

Reviewer 1 Report

The aim of this narrative review is to evaluate the present literature on the effects of exercise on the maintenance of the telomere region and the activation of repair mechanisms in sarcopenia.

The topic is interesting and innovative. The structure of the review is good and the bibliographical references are numerous and cover all the many aspects well.

However, there are some aspects that need to be sorted out before possible publication:

1. Sarcopenia is discussed in the review. However, little attention is paid to sarcopenic obesity, which is equally important. The authors should add a paragaph discussing SO in relation to nutrition and the effects of obesity on the parameters considered (telomere length, genetics etc etc).

2. The nutritional part of the paper is poor. The authors should elaborate on the risks of supplementation, they only highlighted some of the benefits. Some suggestions on proper nutrition should also be included, highlighting the benefits of calorie restriction without malnutrition, and of eating a plant-based diet. 

3. Abastract is very superficial. There is little information compared to what is written in the text. 

4. Paragraph 6 (telomerase) is rather difficult to read. A figure should be included. 

5. All the great amount of information is summarised in obvious information. Correct nutrition and 150 minutes of physical activity per week. Is it possible to amplify the conclusions and take-home messages based on the data presented?

Author Response

We are truly grateful for the comments on the manuscript. These comments were very helpful and enable us to significantly improve the quality of our paper.

The corrections are marked with yellow

Comment

  1. Sarcopenia is discussed in the review. However, little attention is paid to sarcopenic obesity, which is equally important. The authors should add a paragaph discussing SO in relation to nutrition and the effects of obesity on the parameters considered (telomere length, genetics etc etc).

Response

Thank you very much for your suggestion, we have included  in the section 5.2, the missing information on sarcopenic obesity. It was also added to a study in Table 1 on the association of telomeric length and sarcopenic obesity (study conducted by Goddard et al., 2022).

Comment

  1. The nutritional part of the paper is poor. The authors should elaborate on the risks of supplementation, they only highlighted some of the benefits. Some suggestions on proper nutrition should also be included, highlighting the benefits of calorie restriction without malnutrition, and of eating a plant-based diet. 

Response

Thank you very much for your suggestion, the topic on the nutritional role and benefits of caloric restriction without malnutrition was expanded.

Comment

  1. Abastract is very superficial. There is little information compared to what is written in the text. 

Response

Thank you very much for your comment and suggestion. The abstract was changed, with greater emphasis on the beneficial effects of physical activity on telomeric length in sarcopenia and repair mechanisms involved.

Comment

  1. Paragraph 6 (telomerase) is rather difficult to read. A figure should be included. 

Response

Thank you very much for your comment and suggestion. We have included Figure 2,  based on aspects related to telomerase.

Comment

  1. All the great amount of information is summarised in obvious information. Correct nutrition and 150 minutes of physical activity per week. Is it possible to amplify the conclusions and take-home messages based on the data presented?

Response 

Thank you very much for your comment and suggestion. The conclusion was expanded.

Reviewer 2 Report

This is a very insightful review that provides useful information for people when considering healthy aging. The writing at times is very poor and needs to be improved. In numerous instances sentences are incomplete or a new sentence commences while continuing with the thoughts/information stated in the previous sentence. There are numerous abbreviations that need to be checked for consistency throughout the manuscript. Below are further edits and comments.

Line 20: “Consequently, the objective of this review….”

Line 43: “…presenting chronic NCDs…”

Line 45-46: “In this context, the present review aims to present state of the art of knowledge regarding….”

Lines 50-52: This sentence could be better written.

Line 61: “…chronic NCDs..”

Line 73-81: This sentence is too long and needs to be broken up as well as written in a more concise manner.

Line 79-80: “…chronic NCDs..” – please be consistent throughout the whole manuscript with abbreviations.

Line 163: Physiotherapy for “marching”. Not sure what this means. You need to emphasise the importance of resistance training for addressing sarcopenia.

Lines 20-203: This sentence is incomplete.

Line 226: “CRP”

Line 237: “There is strong evidence linking sarcopenia and the inflammatory process …..”

Line 239: “CRP” changes are required throughout the whole manuscript.

Lines 245-246: “In another similar study, an increase in IL-6 and TNF-α was associated with poor exercise habits and nutritional practices in people with sarcopenia.”

Lines 249-253: This sentence is too long and needs to be revised. Please try to write in a more concise and clear manner.

Lines 338-339: This sentence is incomplete.

Table 1: “chair sit-to-stand”; “appendicular skeletal muscle index”

Line 486: “….maintaining telomeres length..”

Line 498: Do you mean “OxS”?

Line 535: Need to correct this sentence so that it is consistent with the ACSM recommendations which state “The ACSM recommends that most adults engage in moderate-intensity cardiorespiratory exercise training for ≥30 min·d on ≥5 d·wk for a total of ≥150 min·wk, vigorous-intensity cardiorespiratory exercise training for ≥20 min·d on ≥3 d·wk (≥75 min·wk), OR a combination of moderate- and vigorous-intensity exercise to achieve a total energy expenditure of ≥500-1000 MET·min·wk.

Line 550: Spell out “FR” before abbreviating.

Line 598: “homeostasis”

Lines 602-607: The sentence is too long, convoluted, and very poorly written. Please revise.

Author Response

We are truly grateful for the comments on the manuscript. These comments were very helpful and enable us to significantly improve the quality of our paper.

Corrections are marked with green

Comments

This is a very insightful review that provides useful information for people when considering healthy aging. The writing at times is very poor and needs to be improved. In numerous instances sentences are incomplete or a new sentence commences while continuing with the thoughts/information stated in the previous sentence. There are numerous abbreviations that need to be checked for consistency throughout the manuscript. Below are further edits and comments.

Line 20: “Consequently, the objective of this review….”

Line 43: “…presenting chronic NCDs…”

Line 45-46: “In this context, the present review aims to present state of the art of knowledge regarding….”

Lines 50-52: This sentence could be better written.

Line 61: “…chronic NCDs..”

Line 73-81: This sentence is too long and needs to be broken up as well as written in a more concise manner.

Line 79-80: “…chronic NCDs..” – please be consistent throughout the whole manuscript with abbreviations.

Line 163: Physiotherapy for “marching”. Not sure what this means. You need to emphasise the importance of resistance training for addressing sarcopenia.

Lines 20-203: This sentence is incomplete.

Line 226: “CRP”

Line 237: “There is strong evidence linking sarcopenia and the inflammatory process …..”

Line 239: “CRP” changes are required throughout the whole manuscript.

Lines 245-246: “In another similar study, an increase in IL-6 and TNF-α was associated with poor exercise habits and nutritional practices in people with sarcopenia.”

Lines 249-253: This sentence is too long and needs to be revised. Please try to write in a more concise and clear manner.

Lines 338-339: This sentence is incomplete.

Table 1: “chair sit-to-stand”; “appendicular skeletal muscle index”

Line 486: “….maintaining telomeres length..”

Line 498: Do you mean “OxS”?

Line 535: Need to correct this sentence so that it is consistent with the ACSM recommendations which state “The ACSM recommends that most adults engage in moderate-intensity cardiorespiratory exercise training for ≥30 min·d on ≥5 d·wk for a total of ≥150 min·wk, vigorous-intensity cardiorespiratory exercise training for ≥20 min·d on ≥3 d·wk (≥75 min·wk), OR a combination of moderate- and vigorous-intensity exercise to achieve a total energy expenditure of ≥500-1000 MET·min·wk.”

Line 550: Spell out “FR” before abbreviating.

Line 598: “homeostasis”

Lines 602-607: The sentence is too long, convoluted, and very poorly written. Please revise.

 Response

Thank you very much for your review, comments and suggestions. We have made all the corrections indicated

Reviewer 3 Report

It is interesting for readers.

Some points to be revised.

-Oxidative stress. It would be needed to add some data from clinical studies.

-In Figure 1, numbers and letters are too small, impairing visibility.

-Telomerase, this section is very comprehensive and based on extensive clinical trial data.

-I don't think the " [Verlaan et al., 2018" description in Line 523 is the correct way to cite it.

-In 8. Sarcopenia prevention interventions, what drug therapies seem promising for improving sarcopenia?

Author Response

We are truly grateful for the comments on the manuscript. These comments were very helpful and enable us to significantly improve the quality of our paper.

Corrections are marked with blue. 

Comment

-Oxidative stress. It would be needed to add some data from clinical studies.

Response

Thank you very much for your suggestion, clinical studies were added in section 4.1.

Comment

-In Figure 1, numbers and letters are too small, impairing visibility.

Response

Figure 1 was modified according to suggestions.

Comment

-Telomerase, this section is very comprehensive and based on extensive clinical trial data.

Response

Thank you very much for your suggestion, it was included Figure 2, on telomerase.

Comment

-I don't think the " [Verlaan et al., 2018" description in Line 523 is the correct way to cite it.

Response

Thank you very much for your comment, we have reviewed and corrected the quotations

Comment

-In 8. Sarcopenia prevention interventions, what drug therapies seem promising for improving sarcopenia?

Response

Thank you very much for your comment, section 8, information on promising drug treatments to improve sarcopenia, was included.

Round 2

Reviewer 1 Report

The authors fixed the paper following my indications

Author Response

Thank you very much for your review and comments.

Reviewer 2 Report

Well done on responding adequately to my comments and suggestions.

Author Response

(The authors gave the same response as above.)
